# Silica Microparticles from Sugarcane By-Products as an Encapsulation System for Retinoids Aimed at Topical Sustained Release

**DOI:** 10.3390/ijms25063215

**Published:** 2024-03-12

**Authors:** Joana R. Costa, Ana Helena Costa, João Azevedo-Silva, Diana Tavares-Valente, Sérgio C. Sousa, Tânia Neto, Manuela E. Pintado, Ana Raquel Madureira

**Affiliations:** 1CBQF—Centro de Biotecnologia e Química Fina–Laboratório Associado, Escola Superior de Biotecnologia, Universidade Católica Portuguesa, Rua de Diogo Botelho 1327, 4169-005 Porto, Portugalsdsousa@ucp.pt (S.C.S.); taneto@ucp.pt (T.N.); mpintado@ucp.pt (M.E.P.); 2Amyris Bio Products Portugal, Rua de Diogo Botelho 1327, 4169-005 Porto, Portugal; jasilva@ucp.pt (J.A.-S.); dvalente@ucp.pt (D.T.-V.)

**Keywords:** retinol, silica microparticles, response surface methodology, control release, topical delivery

## Abstract

The encapsulation of retinol within silica microparticles has emerged as a promising opportunity in the realm of cosmetic and pharmaceutical formulations, driven by the need to reinforce the photoprotection and oxidation stability of retinol. This work examines the process of encapsulating retinol into silica microparticles. The association efficiency, microparticle size, molecular structure, morphology, oxidation, and release profile, as well as biocompatibility and skin sensitization, were evaluated. Results showed that 0.03% of retinol and 9% of emulsifier leads to an association efficiency higher than 99% and a particle size with an average of 5.2 µm. FTIR results indicate that there is an association of retinol with the silica microparticles, and some may be on the surface. Microscopy indicates that when association happens, there is less aggregation of the particles. Oxidation occurs in two different phases, the first related to the retinol on the surface and the second to the associated retinol. In addition, a burst release of up to 3 h (30% free retinol, 17% associated retinol) was observed, as well as a sustained release of 44% of retinol up to 24 h. Encapsulation allowed an increase in the minimal skin cytotoxic concentrations of retinol from 0.04 μg/mL to 1.25 mg/mL without skin sensitization. Overall, retinol is protected when associated with silica microparticles, being safe to use in cosmetics and dermatology.

## 1. Introduction

Vitamin A occurs naturally in several different forms: retinol, retinaldehyde (retinal), retinoic acid (RA), and beta-carotene. Compounds analogous to vitamin A in structure or function are collectively known as retinoids, which include naturally occurring and synthetic compounds [1]. The biologic effects of these retinoids are numerous and include modulation of cellular proliferation and differentiation, anti-keratinization, immunologic effects, tumor prevention and therapy, induction of apoptosis, effects on extracellular matrix components, alteration of cellular cohesiveness, and anti-acne effects [1].

Retinol does not exert a significant biological effect on tissues but becomes active after transformation into more active metabolites, the most important one being retinoic acid, which is characterized by its multilateral action. Two-step oxidation occurring in the target organ cells results in the conversion of retinol to its active form—retinoic acid. After entering the cell, retinol dehydrogenase or alcohol dehydrogenase catalyzes the oxidation of retinol to retinal, and subsequently, retinol is oxidized to retinoid acid by retinaldehyde dehydrogenase [2]. 

Retinoic acid, or tretinoin, is the biologically active retinoid that is involved in producing beneficial effects on the skin, such as the stimulation of collagen production, reducing the appearance of fine lines and wrinkles, improving skin texture, and inhibiting the production of melanin. It is commonly used in the treatment of acne [3,4,5]. In addition, it has a short lifetime due to its degradation by a cytochrome P450-dependent monooxygenase system and induces undesirable side effects (congenital malformations as well as mucocutaneous dryness, headache, and hypertriglyceridemia) when administered at high concentrations [6]. Therefore, several groups have developed different RA delivery systems to overcome these limitations. Delivery systems have been based on polymeric scaffolds (e.g., hydrogels, nanofibers), nanoparticles (liposomes, micelles, polymeric, dendrimers), and microparticles, among others [6]. Table 1 summarizes the major classes of delivery systems described in the literature for lipophilic molecules, including retinoids. 

While retinol is used as a primary active ingredient in many skin care formulations, its efficacy is often limited by an extreme sensitivity to degradation due to light and oxygen susceptibility and is converted to retinoic acid in the formulation, losing its activity. In addition, it is not advisable to add retinoic acid directly to the skin as it is toxic at high concentrations, causing symptoms of irritation including peeling, erythema, and photosensitivity [17]. 

Silicon (Si) is an inorganic elemental, with its oxide configurations of silicate (SiO_4_) and silica (silicon dioxide, SiO_2_) being commonly used in various industries and frequently employed for biomedical applications. The inclusion of amorphous silica in various leave-in cosmetic formulations is widespread, especially in formulations targeting the hair, skin, lips, face, and nails, and improves product efficacy, texture, and extension of shelf-life. Additionally, silica nanoparticles have gathered attention for exhibiting significant photostability and efficacious protection against ultraviolet radiation [18]. Most of the available commercial silica is of sand origin but there is a high demand for sustainable and green sources. 

The exceptional characteristics that SiO_2_ possesses, such as expansive surface area, biocompatibility, porosity, and adaptability for surface modifications, alongside the ability to control parameters such as size and shape, simple large-scale synthesis, and cost-effectiveness, render SiO_2_ microparticles as a highly promising resource for application in the dermatology and cosmetic industries [18]. The encapsulation of substances using a combination of inorganic particles and organic polymers can lead to modifications in the surface attributes of the encapsulated cores. This encapsulation approach has the potential to significantly augment the storage stability of the materials contained within the encapsulated materials [19]. 

A diverse array of retinoids finds application in cosmetics and dermatology, namely oral variants utilized in the management of skin conditions unresponsive to topical interventions, topical formulations including retinol and retinoic acid used for mitigating skin aging, and resources for addressing acne and photo-aging concerns [20]. In addition, other retinoids such as retinaldehyde, adapalene, and tazarotene, similar in function to retinoic acid, are also employed for photo-aging treatment, among many others employed within cosmetic and dermatological contexts [20]. Nevertheless, the use of retinoids in cosmetics also presents several obstacles and side effects, including potential skin irritation and high reactivity to light and oxygen, which challenge their stability in a final product. 

Incorporation of retinoids in topical delivery systems may enhance their stability and bioavailability, through protection from light, temperature, and oxygen, and may also reduce some potential side effects, such as skin irritation, through the controlled release of retinol [17]. Recently, our research group has developed silica particles from sugarcane ash as a sustainable source [21]. Thus, we propose using biogenic silica extracted from sugarcane bagasse ash to produce biocompatible encapsulation systems for retinoids, increasing their stability and skin activity.

## 2. Results

### 2.1. Modelling and Optimization of Retinol-Loaded Silica Microparticles

Table 2 summarizes the results of the complete experimental design regarding association efficiency and particle size. To better visualize the impact of the concentrations of retinol and emulsifier on the association efficiency, the surface response is outlined in Figure 1. The different microparticle formulations allowed us to associate retinol with efficiencies ranging from 62.48 to 99.32%, as shown in Table 2. 

Regarding the particle size, the different formulations yielded microparticles with average sizes ranging from 4.8 to 10.4 μm (Table 2), which are slightly larger values than the average diameters of empty silica microparticles, ca. 3.9 μm [21]. 

Table 3 shows the regression coefficients for the coded polynomial equations and the determination coefficients (R^2^). Some non-significant terms were eliminated, and the resulting equations were tested for adequacy and fitness by ANOVA. The R^2^ values were 0.945 and 0.869 for association efficiency and particle size, respectively, showing a good correlation between the model and the experimental data for both byproducts. 

### 2.2. Structural Characterization

The FTIR spectra of Retinol CB 50 (Figure 2) showed five major peaks at 2925–2850 cm^−1^, corresponding to the C-H asymmetric and symmetric stretching of a methylene group, a C=C stretch at 1740 cm^−1^, a C-H stretch of alkane methyl groups at 1456 cm^−1^, and finally C-O-C and C-O-H stretches between 1160–1094 cm^−1^, corresponding to the stretching of aliphatic ether compounds. Finally, at 720 cm^−1^, the stretching of aliphatic ether compounds and the C=C bending of alkene were disubstituted (cis). The spectra for silica microparticles mainly present a large band between 1240–1040 cm^−1^, which represents the asymmetric stretching vibrations of siloxane covalent bonds [21]. The spectra for retinol-loaded silica microparticles are mainly composed of the 1240–1040 cm^−1^ band of silica, and some peaks that are also present in retinol, such as 2925–2850 cm^−1^ and 1740 cm^−1^.

Scanning Electron Microscope (SEM) images of retinol-loaded silica particles of formulations Fo (optimal theoretical formulation) and F5 (Figure 3) revealed different morphologies. While the theoretically optimal formulation (Fo) presented an agglomeration of silica nanoparticles without a defined shape, the experimental best formulation (F5) allowed the perfect aggregation of silica particles into very spherical microparticles. Moreover, SEM analysis allowed us to confirm that Fo retinol-loaded silica particles presented particle sizes of ca. 2–10 μm, as expected from the surface response (Figure 1b). 

### 2.3. Oxidative Profile and Release

Table 4 presents the results obtained from DSC analyses of retinol and retinol-loaded silica microparticles. An initial assessment of the thermo-oxidative profile of retinol allows us to observe a broad exothermic band with an onset point at 181 °C and end at 380 °C, where major decomposition occurred at 250 °C, similar to other vitamin profiles [22,23]. The thermograms of retinol’s association with the biogenic silica particles show two different peaks, with the first peak occurring at temperatures similar to free retinol, 211.7 °C and 240.6 °C for F5 and Fo, respectively, which is due to the retinol adsorbed at the microparticle surface. Additionally, F5 presents more free retinol in its composition, which justifies the lower onset point and translates into a large broad peak and, therefore, more energy released during the melting phase. The second exothermic peak corresponds to the thermo-oxidation of the encapsulated retinol and occurred at 315–317 °C, with onset temperatures of 273–275 °C.

### 2.4. In Vitro Release of Retinol

The release profile of retinol from silica particles was studied using Fo and F5 formulations as models, containing 30 and 10 mg of retinol, respectively. Both formulations showed an initial burst effect during the first three hours, although this was slightly higher for the optimal formulation (Fo), with approximately 30% of retinol being released in the first three hours (Figure 4). The initial release of retinol from F5 was 17% after 3 h and 37% after 6 h, which was very similar to Fo (35% of retinol released after 6 h). After 24 h, F5 released approximately 44% of the total loaded retinol, while Fo released approximately 55%. 

### 2.5. Biocompatibility

For the purpose of this work, free retinol was considered cytotoxic above 30% of metabolic inhibition. The findings demonstrate that concentrations of free retinol above 0.04 μg/mL exhibit a metabolic inhibition exceeding 30%, thus being cytotoxic to the cells (Figure 5). These results are very different from previous work from Kim et al., who did not find any significant cytotoxicity for concentrations of retinol between 25 to 200 µg/mL [24]. These results could be explained by the type of retinoid and origin, as synthetic retinoids might be able to increase natural retinoid efficacy while reducing their toxicity [6]. As expected, the encapsulation of retinol with both Fo and F5 silica particle formulations enabled us to increase the minimal non-cytotoxic concentration from 0.04 μg/mL to 1.25 mg/mL, allowing us to conclude that a gradual and sustained delivery of retinol was achieved. These results are in accordance with the discussion provided by Singh and Das, who compared the effect of free retinol and liposome-encapsulated retinol on red blood cell membranes and observed that when retinol was incorporated into liposomes, it caused membrane lysis to a lesser degree [25]. 

### 2.6. Skin Sensitization Potential

A summary of DPRA results for retinol-loaded silica microparticles is presented in Table 5. 

The results of the DPRA test method are used to classify chemicals as having minimal, low, moderate, or high reactivity, thus supporting the discrimination between sensitizers and non-sensitizers. Formulations Fo and F5 of retinol-loaded silica microspheres showed similar lysine depletion capacity (0.10 ± 0.01 and 0.21 ± 0.02 %, respectively), but F5 promoted further cysteine depletion (6.74 ± 2.23% vs. 1.69 ± 0.72% of cysteine depletion by Fo). Nevertheless, according to the OECD test method, this result represents a minimal reactivity [26]. 

## 3. Discussion

Retinoids are widely used in cosmetics as an effective dermatological agent and include either natural or synthetic analogs of vitamin A. This work used DSM Retinol CB 50, containing different retinoids, including all-*trans* retinol, 9-*cis* retinoic acid, retinal, and 11-*cis* retinol. 

Formulations of retinol-loaded silica particles for the potential topical delivery of retinol were optimized and studied to understand how the retinol and emulsifier concentrations influence the physical properties of these particles. According to our results, the concentration of emulsifier was the parameter with the highest impact on retinol association efficiency, in line with the work of Shields et al. [17]. They concluded that lower volumes of co-solvent lead to retinol precipitation when added to the reaction, while higher volumes stabilized it. In addition, according to Lee et al., silica particles with an irregular shape were not able to entrap retinol inside the microspheres, indicating that the shape of the material used can interfere with the association efficiency [19]. The impact of the emulsifier and retinol concentrations on particle size followed a quadratic model, with the use of average concentrations allowing us to achieve the largest particle sizes. Modelling and analyzing the surface response for retinol association efficiency allowed us to conclude that the most adequate operational condition, according to the theoretical model (Fo), should contain a retinol concentration of 1.17% and an emulsifier concentration of 17.49. Formulation 5 (F5), containing 0.03% retinol and 9.0% emulsifier, presented the highest association efficiency, and was also evaluated for safety and functionality in this work.

FTIR results may allow us to conclude that retinol might not be completely associated but is instead adsorbed to the silica surface. In addition, there was no formation of new chemical groups. However, microscopy characterization showed that even though some retinol may be adsorbed on the surface of the particles as the FTIR data suggest, that amount is not enough to promote the clustering of the material.

A significant challenge in the formulation of retinol is its vulnerability to degradation upon exposure to light and oxygen. DSC allowed us to confirm the thermal stability of the prepared retinol-loaded silica particles as well as their thermo-oxidative stability under an oxygen atmosphere [17]. Therefore, the encapsulation of retinol in silica microspheres seems a promising approach to enhance its thermal stability, which is important for its use in various applications, including pharmaceuticals and cosmetics.

The achievement of a sustained release of retinol from silica particles is a determining factor in controlling the delivery of retinol to the skin, minimizing sudden exposure and potential irritation, and this was confirmed by in vitro release studies. The release of retinol from silica particles has been studied in the context of controlled delivery systems and the encapsulation efficiency and release of retinol from these particles have been found to be influenced by factors such as the surfactant and its concentration. Our results demonstrated that both formulations showed a rapid initial release of retinol, which can be interpreted by the liberation of molecules situated close to the microsphere surface and by the affinity of retinol towards the release medium. Afterwards, the consistent release of retinol is a result of the penetration of sweat into the particle matrix and the diffusion of retinol through the matrix [27]. Fo showed faster release kinetics initially, which, according to Goudon and colleagues, can be supported by the larger retinol:emulsifier mass ratio of Fo compared to F5 [27]. Moreover, none of the formulations released more than 60% of the total content of retinol in 24 h, suggesting that silica microparticles could be used to adsorb retinol for topical controlled release, but are not very effective as an encapsulation agent. 

While retinol is generally considered safe and effective, there have been discussions regarding its cytotoxic effects. The concern arises from the fact that retinol can induce skin irritation, redness, and peeling, especially when used in higher concentrations or without proper precautions. This irritation may lead to damage to the skin barrier, potentially impacting the viability of skin cells. Some research suggests that high concentrations of retinol can induce oxidative stress, leading to cell damage. Additionally, the potential for irritation and dryness associated with retinol use may exacerbate existing skin conditions, making it crucial for individuals to use retinol products judiciously. While amorphous silica is documented as “Generally Recognized As Safe” (GRAS) by the FDA and has found extensive use in cosmetics as well as being a food additive, it is important to note that data gathered from in vitro toxicity and compatibility assessments are subject to variation based on factors such as cell type, particle size and shape, and potential contaminants [21]. The encapsulation of retinol into silica particles not only shields retinol from external factors that can degrade its stability but also controls its release, reducing the risk of irritation and toxicity [24].

The DPRA is an in chemico test method used to assess the sensitization potential of chemical compounds, particularly their reactivity with skin proteins. It is a validated method for modelling the first key event in the skin sensitization pathway by identifying dermal sensitizers based on their reactivity with synthetic peptides containing the amino acid residues lysine and cysteine. The skin sensitization potential of biogenic silica microparticles was previously assessed through DPRA and allowed us to confirm their safety [28]. Nevertheless, retinol and its derivatives are highly reactive molecules that can lead to significant localized irritation, characterized by slight redness (mild erythema) and the shedding of the outermost layer of the skin. Therefore, it is essential to test the safety of retinol-loaded silica particles [27,28]. Based on the low depletion of cysteine (1.69 ± 0.72% and 6.74 ± 2.23% for Fo and F5, respectively) and the low average depletion of cysteine and lysine (0.21 ± 0.02% and 0.10 ± 0.01% for Fo and F5, respectively), the reactivity of retinol-loaded silica microparticles on these peptides should be considered minimal. Based on these minimal reactivities, either for cysteine or for cysteine and lysine, and according to the DPRA prediction model, retinol-loaded silica microparticles can be classified as non-sensitizers. In vivo skin irritation results obtained by Castro and co-workers detail a considerable improvement of skin tolerability for retinoic acid when encapsulated into solid lipid nanoparticles [29]. In fact, encapsulation presents a promising possibility for addressing the challenges associated with retinol toxicity in skincare by offering a protective shield and controlled release mechanism, enhancing the safety and efficacy of retinol.

Overall, the association of retinol with silica microparticles allows us to reduce the likelihood of surface-level irritation, dryness, and flakiness, making the product more tolerable, especially for individuals with sensitive skin. Additionally, encapsulated retinol creates a more stable product with a prolonged shelf life as it protects the retinol from degradation and increases its stability.

## 4. Materials and Methods

### 4.1. Materials

Sugarcane bagasse ash (SCBA), obtained through the incineration of sugarcane bagasse, was provided by Raízen (Brotas, São Paulo, Brazil). The raw material was sieved, and the fraction with a particle size larger than 160 μm was used for the extraction of biogenic silica. Retinol CB 50, containing ca. 60% retinol, was obtained from DSM (Netherlands). Sulfuric acid (p.a 95.0–97.0%), trifluoroacetic acid (HPLC grade, p.a > 99%), dimethyl sulfoxide (DMSO), and sodium chloride, (p.a > 99.5%) were purchased from Honeywell (Charlotte, NC, USA). Sodium hydroxide pellets were purchased from LabChem (Zelienople, PA, USA); isopropyl alcohol, cinnamic aldehyde, and urea (ACS reagent, 99.0–100.5%) were purchased from Sigma Aldrich (St. Louis, MO, USA). Cosmetic emulsifier EmulpharmaR Core was obtained from Res Pharma Industriale (Trezzo sull’Adda, Italy) and (S)-Lactic acid was purchased from Merck (Darmstadt, Germany). 

Dulbecco’s Modified Eagle’s Medium (DMEM), Fetal Bovine Serum (FBS), and PrestoBlueTM reagents were acquired from Thermo Scientific (Roskilde, Denmark), and Penicillin-Streptomycin (Pen-Strep) from Lonza (Basel, Switzerland). The synthetic peptides containing cysteine (Ac-RFAACAA-COOH; MW = 750.9 g/mol; purity > 95%) or lysine (Ac-RFAAKAA-COOH; MW = 775.9 g/mol; purity > 95%) were obtained from GenScript Biotech (Leiden, The Netherlands).

### 4.2. Synthesis of Retinol-Loaded Silica Microparticles

#### 4.2.1. Synthesis of Silica Gel

Silica microspheres were produced from sugarcane bagasse ashes, according to Costa et al., with slight modifications [21]. Briefly, nickel-plated stainless steel crucibles were used to mix ash and sodium hydroxide and placed into a muffle oven at 350 °C for 20 min. After the removal of the crucibles, deionized water was then added, and the mixture was thoroughly stirred and filtered through a paper filter. The solid fraction was discarded and the liquid fraction, rich in sodium silicate, was adjusted to pH 6 using H_2_SO_4_ 10% (*v*/*v*) under agitation. The solution was then filtered under vacuum, using a cloth filter, and thoroughly washed with deionized water. The final gel was resuspended in water and the viscosity of the silica gel solution was adjusted to 10–12 cP using a Vibro Viscometer SV-10 (A&D Co., Tokyo, Japan).

#### 4.2.2. Incorporation of Retinol Extract

Retinol, at different concentrations, was added to the emulsifier and homogenized through ultrasonication (CY-500 Optic Ivymen System) for 3 min with a wave amplitude of 75%. The mixture was then added to the solution of silica gel and homogenized again for 3 min. The sample was left stirring overnight at room temperature and protected from light, before being dried in a mini spray-dryer (Buchi Mini Spray B-290, Buchi, Lucerne, Switzerland). Samples were dried using an inlet temperature of 120 °C, pump 13%, and aspirator rate 65%.

### 4.3. Experimental Design

The best parameters were determined by response surface methodology according to a 2^2^ central composite design, using the association efficiency of retinol as the response. Two factors were analyzed as independent variables, namely the retinol concentration and the emulsifier concentration, and both were evaluated at five levels according to Table 6. The following polynomial equation was fitted to data:y = β_0_ + β_1_x_1_ + β_2_x_2_ + β_11_x^2^_1_ + β_22_x^2^_2_ + β_12_x_1_x_2_,(1)
where βn are constant regression coefficients, y is the response (association efficiency), and x_1_ and x_2_ are the coded independent variables (retinol and emulsifier concentrations, respectively).

### 4.4. Association Efficiency of Retinol

Quantification of associated retinol was determined according to the method described by Shields IV et al. [17]. Briefly, in this assay, an ethanolic solution containing the spray-dried particles was centrifuged at 4696× *g* for 10 min. The supernatant was discarded, and the pellet was suspended in isopropyl alcohol and centrifuged at 20,000× *g* for 10 min. The process was repeated twice. The supernatant was then diluted and measured by a UV–Vis spectrophotometer. By fitting the spectroscopy data of known concentrations of retinol with the Beer–Lambert Equation, we established a calibration curve and found that the molar absorption coefficient, ε, for retinol in isopropyl alcohol at 450 nm was 38,163. This data and the known volume of liquid in the solution allowed us to determine the mass of associated retinol.

### 4.5. Structural Characterization

Particle size distribution was assessed with a Mastersizer 3000E equipped with a Hydro EV sample dispersion unit (Malvern Panalytical, Malvern, UK). Six measurements were performed, using the samples dispersed in water at 1 wt%, with results expressed as a number distribution. Average particle size, Dx (50), and Dx (90) results were assessed.

Scanning electron microscopy microparticle analyses were performed on a Phenom XL G2 (Thermo Fischer Scientific, Eindhoven, The Netherlands). Samples were placed on the top of metallic pins covered with double-sided adhesive carbon tape (NEM tape, from Nisshin, Japan) and coated with gold/palladium using a Sputter Coater (Polaron, from Bad Schwalbach, Germany). All observations were performed (and micrographs acquired) in a high vacuum, with an acceleration voltage of 5 kV, using the secondary electron detector (SED).

Analysis by Fourier transformation infrared (FTIR) was performed to identify potential physicochemical interactions between retinol and silica. ATR-FTIR analysis was conducted in the absorbance range of 4000 to 400 cm^−1^ using a Spectrum 100 FTIR spectrometer, (Perkin Elmer, Hopkinton, MA, USA) equipped with an attenuated total reflectance (ATR) accessory (PIKE Technologies, Fitchburg, WI, USA) with a diamond/Se crystal plate.

### 4.6. Evaluation of Oxidative Stability by Differential Scanning Calorimeter (DSC)

Oxidative stability of the associated retinol was determined by a DSC 204 F1 Phoenix (Netzsch, Selb, Germany) Differential Scanning Calorimeter (DSC) with Proteus Analysis software version 7.1.0 (Netzsch, Selb, Germany). The equipment was calibrated with a high-purity indium standard and microparticles samples of approximately 5 mg were weighed into pierced close aluminum pans and placed in the equipment’s sample chamber. An empty pierced closed pan was used as a reference. Experiments were performed under an oxygen flow of 50 mL/min and a nitrogen flow of 40 mL/min, at a heating rate of 10 °C/min. Oxidation temperature (onset temperature and peak temperature) and enthalpy were measured.

### 4.7. In vitro Study of Retinol Release

Lyophilized microparticles (250 mg) were dissolved in the 5 mL of sweat simulation fluid, composed of 1.08% (*w*/*v*) sodium chloride, 0.12% lactic acid, and 0.13% urea, with pH adjusted to 6.5 [30]. Samples were incubated at 32 °C and stirred at 100 rpm in an incubator shaker Innova 40 (New Brunswick Scientific Inc., Edison, NJ, USA). At predetermined time intervals (0, 3, 6, and 24 h), samples were withdrawn, centrifuged at 4696× *g* for 30 min (Megafuge™ 16, Thermo Fisher Scientific), and the supernatant was collected for quantification of free retinol, as described above.

### 4.8. Biocompatibility

#### 4.8.1. Cell Lines and Growth Conditions

HaCaT cells (an immortalized keratinocyte cell line established from adult human skin cells) were obtained from CLS Cell Lines Service (reference 300493, Eppelheim, Germany). The cells were grown using high-glucose DMEM, supplemented with 10% heat inactivated FBS and 1% (*v*/*v*) Pen-Strep. All cells were incubated at 37 °C in a humidified atmosphere with 5% CO_2_.

#### 4.8.2. Cell Viability Determination

Cells were seeded in a 96-well plate (Nucleon Delta Surface, Thermo Scientific, Roskilde, Denmark) at a density of 1 × 10^5^ cells/mL in a volume of 100 μL. After 24 h, the culture medium was then carefully replaced with the different test solutions and incubated in the dark. The sterile retinol was tested at concentrations ranging from 0.003 to 0.08 μg/mL and sterile retinol-loaded silica microparticles were tested at concentrations ranging from 0.15 to 5 mg/mL by directly re-suspending the molecules into cell culture media and heating under agitation until fully solubilized. Plain culture media was used as a positive control and cells with 10% DMSO were used as a negative control. Following 24 h of exposure, 10 μL of PrestoBlue™ were added and the plates incubated for 1 h. Fluorescence at an excitation wavelength of 560 nm and an emission of 590 nm was then measured using a microplate reader (BioTek Synergy, Thermo Scientific, Denmark). Four replicates were used for each concentration tested. Results are expressed as a metabolic inhibition percentage, with an inhibition superior to 30% being considered cytotoxic in accordance with ISO standards [31].

### 4.9. Direct Reactivity Peptide Assay

Skin sensitization potential was assessed through a Direct Reactivity Peptide Assay (DPRA), following OECD Test guidelines [26]. Retinol-loaded silica particles were suspended in acetonitrile (final concentration 100 mM), followed by mixing at 40 °C for 30 min, a sonication bath for 15 min, and being left under agitation at room temperature. For the assay execution, the cysteine model peptide was prepared as a 0.667 mM stock solution in 100 mM phosphate buffer (pH 7.5) and the lysine model peptide was prepared as a 100 mM stock solution in ammonium acetate buffer (pH 10.2). The silica suspension was then incubated for 24 h at room temperature with cysteine or lysine peptides at ratios of 1:10 or 1:50 (*v*:*v*), respectively. Cinnamic aldehyde 100 mM in acetonitrile was used as a positive control. Additionally, a co-elution control without peptide (silica suspension plus peptide buffer) was performed to detect possible interference of silica particles with the peptides. 

Analysis of free peptides was performed using a high-performance liquid chromatography reverse-phase HPLC (Agilent 1260 Infinity II, Agilent Technologies, Santa Clara, CA, USA) attached to a diode array detector (Agilent 1260 DAD HS) and a Poroshell 120 EC C18 column (30 × 150 mm; 2.7 µm). A gradient analysis was carried out with mobile phase A (0.1% TFA in water) and phase B (0.085%TFA in acetonitrile) starting at time 0 with 10% B, going to 25% B in 15 min, 90% B in 3 min, 10% B in 1 min, and kept for 6 min at 10% B. Detection was performed at 220 nm, with a flow rate of 0.35 mL min^−1^, column temperature of 30 °C, and a sample injection volume of 5 μL. Peptide quantification was achieved through the calibration curves of cysteine- and lysine-enriched peptides ranging between 0.033 to 0.528 mM.

### 4.10. Statistical Analysis

For the central composite design, the analysis of variance (ANOVA) tests for lack of fit, determination of the regression coefficients, and the generation of surface responses were carried out using Statistica 7.0 software (StatSoft, Tulsa, OK, USA).

## 5. Conclusions

This work studied the potential of biogenic silica microparticles as a topical delivery system for retinoids. The encapsulation of retinol into amorphous silica microspheres allowed association efficiencies between 62 and 99%, resulting in microparticles displaying a size distribution ranging from 4.8 to 10.4 µm. FTIR analysis supported the successful association of retinol with the microparticles and suggested a possible presence of retinol on the particle surface. SEM images revealed that upon association, optimal formulation presented an agglomeration of silica nanoparticles into a near-spherical shape. Regarding protection against oxidation, amorphous silica exhibited a good performance in protecting retinol, allowing protection from temperatures up to 317 °C. The release profile of retinol from silica spheres exhibited an initial burst release within the first 3 h, followed by a sustained release of ca. 44% within 24 h. 

A skin biocompatibility assessment revealed that free retinol demonstrated cytotoxicity at concentrations above 0.04 μg/mL, leading to a metabolic inhibition of HaCaT cells surpassing 30%. Meanwhile, the association within silica spheres allowed us to increase the active concentrations of retinol, as it only demonstrated toxicity above 1.25 mg/mL. Evaluation of skin sensitization demonstrated that retinol, when associated with silica microparticles, exhibited a lack of sensitization potential for concentrations of 100 mM.

Overall, it was possible to develop silica microspheres able to encapsulate and absorb lipophilic molecules, widely used in dermatology and cosmetics, creating a more stable ingredient with a prolonged shelf-life that was protected from oxygen and light and able to minimize some skin side effects such as skin irritation.

## Figures and Tables

**Figure 1 ijms-25-03215-f001:**
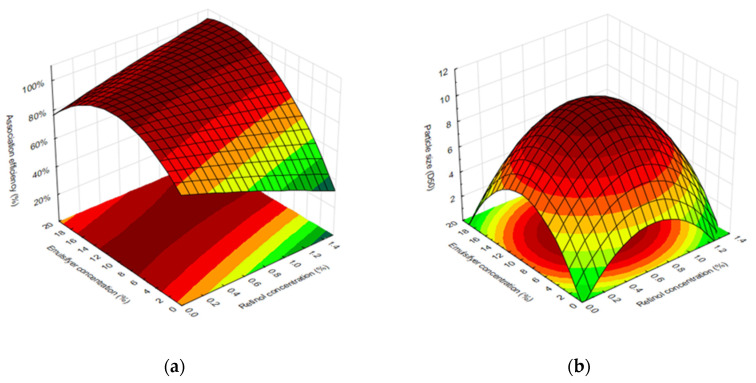
Surface response for (**a**) association efficiency of Retinol CB 50 within biogenic silica microparticles and (**b**) particle size.

**Figure 2 ijms-25-03215-f002:**
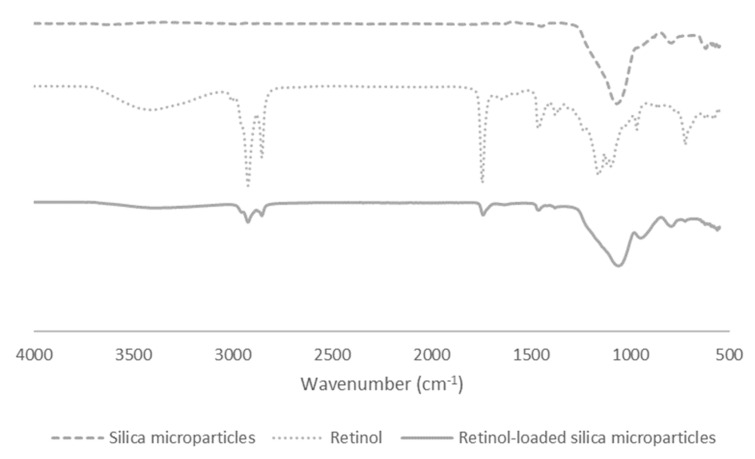
FTIR spectra of biogenic silica microparticles, Retinol CB 50, and retinol-loaded silica microparticles.

**Figure 3 ijms-25-03215-f003:**
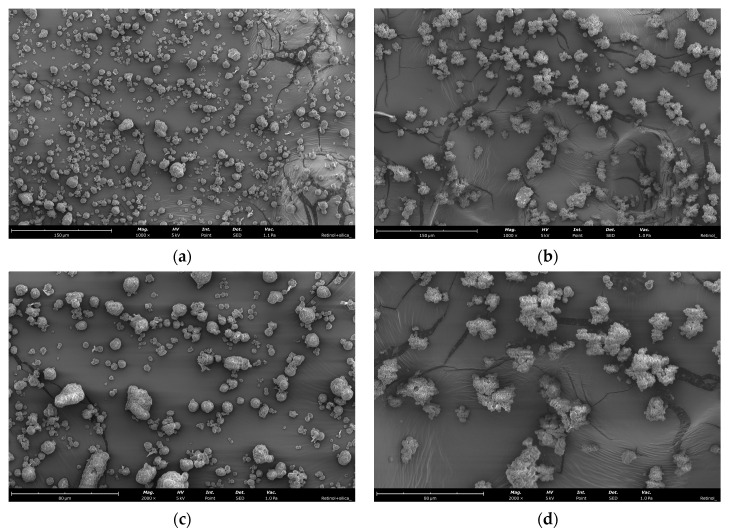
SEM images of (**a**,**c**) Fo and (**b**,**d**) F5 retinol-associated in silica microparticles.

**Figure 4 ijms-25-03215-f004:**
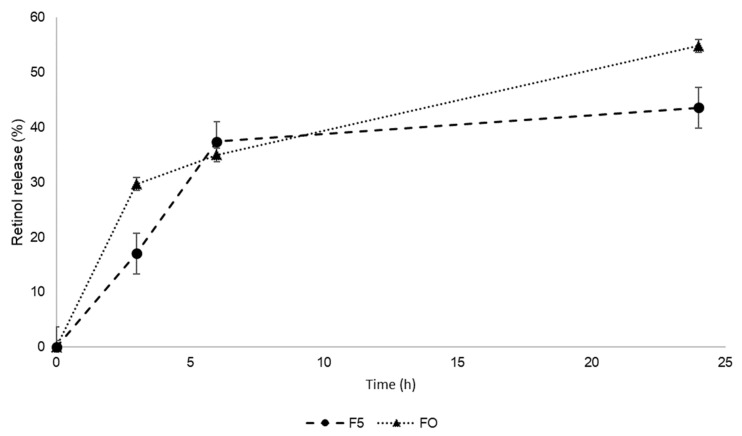
Release profile of Retinol CB 50 (%) from silica microspheres in sweat simulation fluid.

**Figure 5 ijms-25-03215-f005:**
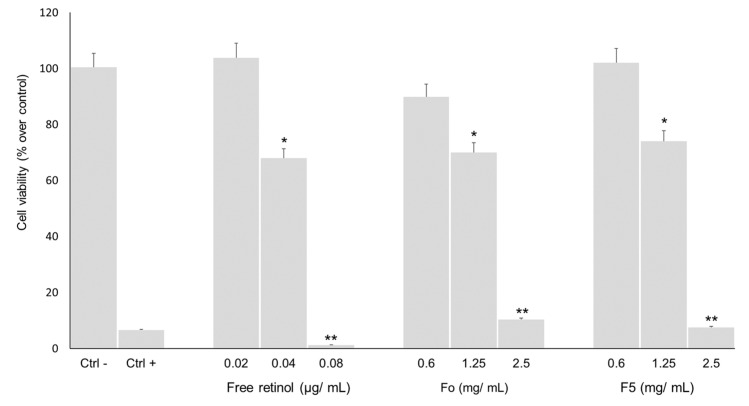
Metabolic inhibition of free and encapsulated Retinol CB 50 (Fo and F5) of HaCaT cells. Values show Mean ± SD (N ≥ 3); * *p* < 0.05; ** *p* < 0.001 vs. control with *p*-values obtained using a two-way analysis of variance (ANOVA) with Tukey’s post-hoc test.

**Table 1 ijms-25-03215-t001:** Description of different topical delivery systems for retinol.

Delivery System	Description	Reference
Solid Lipid Nanoparticles	Non-toxic and biodegradable hydrophilic shell with a hydrophobic core that allows the encapsulation and controlled release of retinoids. Nano-sized particles (82–408 nm). Entrapment efficiency of 78–86%	[7,8,9,10,11]
Nanostructured lipid carriers	Nano-sized colloidal drug delivery system (~100 nm). Biocompatible, biodegradable, non-immunogenic, with high drug loading capacity. Entrapment efficiency of retinol of 92%.	[7,9]
Liposomes	Lipid-based delivery systems with phospholipid bilayers surrounding an aqueous phase. Entrapment efficiency of retinol ranges from 65–97%. Particle sizes > 500 nm.	[8,12]
Niosomes	Nanosized lipid-based delivery systems with a bilayer structure composed of non-ionic surfactants, cholesterol, and phospholipids. Entrapment efficiency of retinol of 92%.	[8,13]
Cyclodextrins	Solubilization of poorly soluble molecules and protection from chemical degradation. Nanometric diameters (<100 nm) and encapsulation efficiencies ranging from 20–50%	[14]
Polymeric micelles	Copolymers of hydrophilic and hydrophobic monomer units (block copolymer). In aqueous media, the hydrophobic portion forms the core, while the hydrophilic portion forms the shell. Nanometric dimensions (10–20 nm and encapsulation efficiencies ranging from 35–80%	[15,16]

**Table 2 ijms-25-03215-t002:** Operational conditions expressed as dimensional independent variables and experimental results obtained for dependent variables, including association efficiency and particle size.

Run	Retinol CB 50 Concentration (%)	Emulsifier Concentration (%)	Association Efficiency (%)	Particle Size (Dx (50))
1	0.20 (−1)	3.00 (−1)	89.81	4.83
2	0.20 (−1)	15.00 (+1)	98.50	7.01
3	1.00 (+1)	3.00 (−1)	66.42	4.85
4	1.00 (+1)	15.00 (+1)	98.97	7.75
5	0.03 (−1.41)	9.00 (0)	99.32	5.23
6	1.17 (+1.41)	9.00 (0)	98.17	7.83
7	0.60 (0)	0.51 (−1.41)	62.48	5.05
8	0.60 (0)	17.49 (+1.41)	98.71	5.44
9	0.60 (0)	9.00 (0)	98.79	10.4
10	0.60 (0)	9.00 (0)	98.84	9.79
11	0.60 (0)	9.00 (0)	98.97	9.42

**Table 3 ijms-25-03215-t003:** Coded second-order regression coefficients for association efficiency.

Coefficient	Association Efficiency	Particle Size
*β* _0_	0.7641	−0.3232
*β*_1_x_1_	−0.2586	12.8194
*β*_2_x_2_	0.0523	1.2008
*β*_11_x^2^_1_	−0.0251	−10.0898
β_22_x^2^_2_	−0.0026	−0.0627
*β* _12_ *x* _1_ *x* _2_	0.0238	0.075
R^2^	0.964	0.869

**Table 4 ijms-25-03215-t004:** Onset temperature, peak temperature, and energy in oxidation of free and associated retinol (F5 and Fo).

	Peak 1			Peak 2		
	Onset (°C)	Peak (°C)	Energy (J/g)	Onset (°C)	Peak (°C)	Energy (J/g)
Retinol CB 50	181	250.8	2200	-	-	-
F5	193.5	211.7	8496	275.6	315.1	1245
Fo	239.0	240.6	2638	273.1	317.3	4554

**Table 5 ijms-25-03215-t005:** Percentage of depletion of cysteine and lysine (mean ± SD) upon incubation with retinol-loaded silica microparticles at 1 mg/mL or cinnamic aldehyde (positive control), and classification of skin reactivity according to the OECD guidelines.

	Cysteine Depletion (%)	Lysine Depletion (%)	Cys and Lys Mean Depletion (%)	Reactivity (cys)	Reactivity (cys and lys)
**Cinnamic aldehyde**	69.36 ± 1.30	53.80 ± 0.88	61.6	Moderate	High
**F5**	6.74 ± 2.23	0.10 ± 0.01	3.3	Minimal	Minimal
**Fo**	1.69 ± 0.72	0.21 ± 0.02	0.4	Minimal	Minimal

**Table 6 ijms-25-03215-t006:** Experimental variables involved in the study.

Coded Variables	−1.41	−1.0	0	+1.0	+1.41
Retinol concentration (%)	0.03	0.2	0.6	1.0	1.17
Emulsifier concentration (%)	0.51	3.0	9.0	15.0	17.49

## Data Availability

Additional data supporting the findings of this study are available from the corresponding author upon reasonable request.

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
