# Peer review of "Silica Microparticles from Sugarcane By-Products as an Encapsulation System for Retinoids Aimed at Topical Sustained Release"

_ijms, 2024, doi:10.3390/ijms25063215_

Round 1
Reviewer 1 Report
Comments and Suggestions for Authors
The manuscript regards the encapsulation of retinol in silica microparticles produced following the procedure previously developed by the authors. The optimized retinol-loaded microparticles in terms of particle size, oxidative profile, in vitro release and biocompatibility have been characterized.
The article meets the scope of the journal, but in my opinion needs major revision before publication.
In Table 1, the authors described the composition of many retinol-loaded microparticles, however in the structural characterization, in tables 3 and 4 only F0 and F5 are mentioned. For a better understanding of the experimental data, I suggest to describe their composition in the results section as well as in the discussion.
About structural characterization, Figure 2 depicted a superimposition of silica microparticles, retinol and loaded microparticles. In materials section the use of retinol at 60% concentration in sunflower oil is reported; which was the retinol employed for FTIR and DSC analyses? It would be better to include an explanation of how the retinol standard was obtained or to indicate in the materials section where it was purchased and its purity degree.
I am unclear about the description of retinol FT IR spectrum: the band at 1740 cm-1 is attributed to C=O stretching of cyclopentanone, but in retinol structure there is not this moiety; furthermore, is cited a C=C bending at 720 cm-1 which is attributed to a cis double bond geometry, but retinol owns an all trans (E) configuration. A C-O stretching typical of aliphatic ether compound is also reported in the description, but no ether bonds are present in retinol structure.
For the oxidative profile study DSC was employed: the retinol behaviour is compared with that of F0 and F5 formulations; in table 3 experimental results are summarized. In literature is known that retinol melting point is at about 60-65 °C, while in table is reported an onset temperature of 181 °C with a peak at 250.8 °C. If retinol standard was used for the analysis, this difference must be justified.
I suggest a thorough review of analytical results and discussion. Furthermore, it would seem necessary to verify if the ultrasonication treatment for retinol incorporation does not cause changes in its structure and geometry.
In addition:
- more attention should be paid in the text to the use of superscripts and subscripts (e.g. Line 62, 63, 72: SiO4 and SiO2; Line 272, Line 319 cm-1).
- The use of F0 instead of Fo is encourage
- Line 141: free retinol 211.7 °C
Comments on the Quality of English LanguageEnglish needs minor revisions throughout the paper
Author Response
We would like to thank the reviewer for his valuable comments and inputs, which contributed to enrich and improve our manuscript. Please find the point-by-point responses below (in blue color).
In Table 1, the authors described the composition of many retinol-loaded microparticles, however in the structural characterization, in tables 3 and 4 only F0 and F5 are mentioned. For a better understanding of the experimental data, I suggest to describe their composition in the results section as well as in the discussion.
We would like to thank the reviewer for the valuable comment. In fact, we tried to explain it in the Discussion section (lines 211 – 214) and added the information regarding F5. Table 1 (Now Table 2) refers to the formulation that were developed under an experimental design – and were characterized regarding association efficiency and particle size. After selecting the best formulations (experimental – F5 and empirical – Fo), these were more further characterized but not the other developed during the optimization step.
About structural characterization, Figure 2 depicted a superimposition of silica microparticles, retinol and loaded microparticles. In materials section the use of retinol at 60% concentration in sunflower oil is reported; which was the retinol employed for FTIR and DSC analyses? It would be better to include an explanation of how the retinol standard was obtained or to indicate in the materials section where it was purchased and its purity degree.
This information was added to materials section (line 304) and the qualitative composition detailed in the discussion section (lines 212 - 214).
I am unclear about the description of retinol FT IR spectrum: the band at 1740 cm-1 is attributed to C=O stretching of cyclopentanone, but in retinol structure there is not this moiety; furthermore, is cited a C=C bending at 720 cm-1 which is attributed to a cis double bond geometry, but retinol owns an all trans (E) configuration. A C-O stretching typical of aliphatic ether compound is also reported in the description, but no ether bonds are present in retinol structure.
We would like to thank the reviewer for the valuable comment. The FTIR discussion section was re-written. Regarding to the C=C bending related with the cis configuration, is related with one of the cis molecules (9-cis retinoic acid or 11-cis retinol) present in the retinol sample used in this work, as described in lines 212 - 214.
For the oxidative profile study DSC was employed: the retinol behaviour is compared with that of F0 and F5 formulations; in table 3 experimental results are summarized. In literature is known that retinol melting point is at about 60-65 °C, while in table is reported an onset temperature of 181 °C with a peak at 250.8 °C. If retinol standard was used for the analysis, this difference must be justified.
In fact, as we used a commercial sample of retinol, containing a mixture of retinoids and solvent, the results are different. However, it seems that our experimentally determined meting point of retinol (181 °C) is in the line with Retinol CB 50 Safety Data Sheet, which described a melting temperature of 196 °C.
I suggest a thorough review of analytical results and discussion. Furthermore, it would seem necessary to verify if the ultrasonication treatment for retinol incorporation does not cause changes in its structure and geometry.
We would like to thank the reviewer for the valuable comment, and the document was all revised, incorporating reviewer’s suggestions and a major revision of results and discussion sections. Regarding the ultrasonication, it is a widely described methodology in the preparation of formulations (solid lipid nanoparticles, liposomes, …) containing retinol that does not affect its structure, due to the low energy that it is used:
Lin, Y., Hsiao, C., Alshetaili, A., Aljuffali, I. A., Chen, E., Fang, J. Lipid-based nanoformulation optimization for achieving cutaneous targeting: Niosomes as the potential candidates to fulfill this aim. Eur. J. Pharm. Sci. 2023. 186
Das, S., Ng, W. K., Kanaujia, P., Kim, S., Tan, R. B. H. Formulation design, preparation and physicochemical characterizations of solid lipid nanoparticles containing a hydrophobic drug: Effects of process variables. Colloids Surf. B: Biointerfaces. 2011, 88 (1), pp. 483-489.
Millar, J., Kelly, H., Payne, C. All-trans retinoic acid solid lipid nanoparticle characterisation and assessment of an-ti-inflammatory activity in a human lung epithelial cell line. RCSI Medical Journal. 2017, 1, pp. 37-43.
Raza, K., Singh, B., Lohan, S., Sharma, G., Negi, P., Yachha, Y., Katare, O. P. Nano-lipoidal carriers of tretinoin with enhanced percutaneous absorption, photostability, biocompatibility and anti-psoriatic activity. Int. J. Pharm. 2013, 456 (1), pp. 65-72.
In addition:
- more attention should be paid in the text to the use of superscripts and subscripts (e.g. Line 62, 63, 72: SiO4 and SiO2; Line 272, Line 319 cm-1). These typos have been corrected.
- The use of F0 instead of Fo is encourage.
We understand the reviwer’s concern, we let explicit that Fo stands for Optimal formulation (line 141).
- Line 141: free retinol 211.7 °C. These typos have been corrected.
Reviewer 2 Report
Comments and Suggestions for Authors
I am attaching all my questions and comments in the file and below.
Review of Silica microparticles from sugarcane by-products as encapsulation system for retinoids aiming topical sustained release
Introduction and Abstract
1. Both elements of the publication introduce the reader well to the topic of research. However, almost half of the literature cited by the authors is over 20 years old - 5 out of 11 cited works. It would be worth using more recent literature on the topic.
2. The Introduction would also benefit from adding an appropriate figure or table regarding the described research/mechanisms.
3. Line 53: Please complete the text with what the RA abbreviation means. I haven't found it anywhere, the reader can only guess what it means.
Materials and methods
1. Line 351-353: “The sterile retinol was tested at concentrations ranging from 0.003 to 0.08 μg/ mL and sterile retinol-loaded silica microparticles were tested at concentrations ranging from 0.15 to 5 mg/ mL.” - is the concentration range of 0.15 - 5 mg/ml the concentration of retinol in microparticles or the microparticles themselves?
2. Cell viability determination section: How were cells treated with retinol alone? Prepared as a solution? In what solvent? I have the same question about microparticles with retinol.
3. 4.8.2. Cell viability determination: What was the positive control? Pure retinol, in what form? Another compound/molecule? At what concentration?
4. Has the influence of the emulsifier been taken into account in studies on the toxicity of microparticles?
Results
1. 2.3. Oxidative profile and release section: What is the reason for the large difference between F5 and Fo samples for first peak for free retinol? Compared to retinol itself according to the data in table 3. Has anyone published similar data before?
2. Fig. 3: The scale in SEM photos is practically illegible. This makes it impossible for the reader to estimate the size of the microparticles.
3. What sizes of microparticles were determined by SEM? Were they comparable to the model?
4. Fig. 2: It would be better if the FTIR spectra were in different colors and had a different spectrum line (e.g. dashed line, solid line, double line). Then a legend is necessary for the figure or in the description of the figure. This makes it easier for people with various vision disorders.
5. 2.4. In vitro release of retinol section: did the study include free retinol or only the adsorbed and encapsulated retinol?
6. Has the efficiency of encapsulation in microparticles been determined? How? What was the efficiency of retinol adsorption on the surface of microparticles?
7. How long does it take for all the retinol to be released? Has this time been specified?
8. 2.5. Biocompatibility: I am curious about the authors' opinion, where could such a large discrepancy in the cytotoxicity of free retinol between their results and the cited results of Kim et al. come from? Citation 23.
9. Have cell survival tests been performed for periods longer than 24 hours?
10. 2.6. Skin sensitization potential: Please add at least a short description of the results.
11. 2.6. Skin sensitization potential: Has the sensitization potential of silica gel and the emulsifier been checked?
Discussion
1. The discussion of the results, due to the volume and various aspects of the research conducted, could be slightly more extensive based on the latest literature.
Conclusions
1. The conclusions are clearly defined based on the research conducted. This part of the work could be supplemented with further perspectives on research and use of the developed microcarrier.
Comments on the Quality of English LanguageThe work is written in very good language. It is worth checking for minor typos and editorial imperfections, but they do not in any way negatively affect the scientific value of the manuscript.
Author Response
We would like to thank the reviewer for his valuable comments and inputs, which contributed to enrich and improve our manuscript. Please find the point-by-point responses below (in blue color).
- Both elements of the publication introduce the reader well to the topic of research. However, almost half of the literature cited by the authors is over 20 years old - 5 out of 11 cited works. It would be worth using more recent literature on the topic.
We thank the reviewer for the valuable observation, we replaced references 3, 4 and 9 for more recent references as suggested. Nevertheless, we need to take into consideration that these references are regarding the biological skin functions of retinoids, which have been described in literature for decades.
The Introduction would also benefit from adding an appropriate figure or table regarding the described research/mechanisms.
This information was added to introduction section (Table 1).
Line 53: Please complete the text with what the RA abbreviation means. I haven't found it anywhere, the reader can only guess what it means.
We would like to thank the Reviewer for alerting us to this typo. The abbreviation meaning was added to line 33.
Materials and methods
- Line 351-353: “The sterile retinol was tested at concentrations ranging from 0.003 to 0.08 μg/ mL and sterile retinol-loaded silica microparticles were tested at concentrations ranging from 0.15 to 5 mg/ mL.” - is the concentration range of 0.15 - 5 mg/ml the concentration of retinol in microparticles or the microparticles themselves?
In fact, sterile retinol-loaded silica microparticles were tested at concentrations ranging from 0.15 to 5 mg/ mL of particle powder, corresponding to 0.045 – 1.5 μg (for formulation F5) and to 1.75 μg – 0.06 mg (formulation Fo).
Cell viability determination section: How were cells treated with retinol alone? Prepared as a solution? In what solvent? I have the same question about microparticles with retinol.
We thank the reviewer for the valuable observation. The required information was added to the methodology section (lines 409 – 410).
Cell viability determination: What was the positive control? Pure retinol, in what form? Another compound/molecule? At what concentration?
We would like to thank the Reviewer for alerting us to this typo. In fact, the positive control is only plain media to guarantee that cells are normally growing (line 410).
Has the influence of the emulsifier been taken into account in studies on the toxicity of microparticles?
We thank the reviewer for the valuable observation. The toxicity of the emulsifier alone was not tested because Emulpharma® CORE is a commercial ingredient for cosmetic use, recommended to be used at concentrations up to 30%, so not toxicity should be expected.
Results
- 3. Oxidative profile and release section: What is the reason for the large difference between F5 and Fo samples for first peak for free retinol? Compared to retinol itself according to the data in table 3. Has anyone published similar data before?
We would like to thank the Reviewer for the valuable input. We added this discussion to the manuscript (Results section).
- 3: The scale in SEM photos is practically illegible. This makes it impossible for the reader to estimate the size of the microparticles.
Figure 3 was uploaded with better definition.
- What sizes of microparticles were determined by SEM? Were they comparable to the model?
We would like to thank the Reviewer for the valuable observation, which was added to the manuscript (lines 144 – 146).
- 2: It would be better if the FTIR spectra were in different colors and had a different spectrum line (e.g. dashed line, solid line, double line). Then a legend is necessary for the figure or in the description of the figure. This makes it easier for people with various vision disorders
Figure 2 was changed according to the reviewer’s suggestions.
- 4. In vitro release of retinol section: did the study include free retinol or only the adsorbed and encapsulated retinol?
Thank you for your comment. The in vitro release study includes the free retinol, as it represents less than 1% of the total retinol, it is not a significant quantity.
- Has the efficiency of encapsulation in microparticles been determined? How? What was the efficiency of retinol adsorption on the surface of microparticles?
We have determined the association efficiency of retinol, as described in section 4.4 (lines 351 – 360), which includes the encapsulated and surface-adsorbed retinol.
- How long does it take for all the retinol to be released? Has this time been specified?
There was no release of the total amount of retinol during the 24 hours of the study, we added this information to the discussion section (lines 254– 257).
- 5. Biocompatibility: I am curious about the authors' opinion, where could such a large discrepancy in the cytotoxicity of free retinol between their results and the cited results of Kim et al. come from? Citation 23.
This reference was altered, as it is referring to Kim et al. A potential explanation is related with the type of retinoid used in the study and if it is either of natural of synthetic origin, which can influence its biological characteristics.
- Have cell survival tests been performed for periods longer than 24 hours?
No, according to ISO 10993-5:2009 test method, it is only required to test for 24 h.
- 6. Skin sensitization potential: Please add at least a short description of the results.
The information required was added to the manuscript. (lines 202 - 208).
- 6. Skin sensitization potential: Has the sensitization potential of silica gel and the emulsifier been checked?
We thank the reviewer for the valuable observation. It is described in section 3 that the skin sensitization potential of silica gel was previously assessed, as described (lines 230 – 231). The sensitization of the emulsifier alone was not tested because Emulpharma® CORE is a commercial ingredient for cosmetic use, recommended to be used at concentrations up to 30%, so it is not expected to present any sensitization.
Discussion and Conclusions
- The discussion of the results, due to the volume and various aspects of the research conducted, could be slightly more extensive based on the latest literature.
- The conclusions are clearly defined based on the research conducted. This part of the work could be supplemented with further perspectives on research and use of the developed microcarrier.
We would like to thank the reviewer for the valuable comments, and the document was all revised, incorporating reviewer’s suggestions and a major revision of results and discussion sections.
Round 2
Reviewer 1 Report
Comments and Suggestions for Authors
Thank you for considering the suggestions given to you.
However, if the retinol standard was not used for analytical data correctness, the name of the preparation used as the standard should be indicated in the text, figures, and captions. Additional components affect and sometimes mask the analytical data of retinol 98% (melting point, trans double bonds etc.) which possesses all-trans configuration of double bonds and a melting point of 62-64 °C
In this revised version, two retinol preparations are indicated in the materials: Retinol CB 50, containing ca. 60% retinol, from DSM (Netherlands) and Retinol from Amyris Inc. at 60% concentration in sunflower oil. It is necessary to specify which one was used as the reference and for silica microparticles because they have different components.
In the previous review I was asking about ultrasonication stability because it is known to cause heating of solution and this factor can affect the stability of retinol.
Author Response
Thank you for your insightful comments and feedback on our manuscript, we appreciate the time and effort you have dedicated to reviewing our work. We agree with your points regarding the need for further clarification on the retinol origin/ composition, which we hope to make clear on materials section (line 305) and in all the captions along the manuscript.
Regarding the ultrasonication, we appreciate the clearing up. In fact, as the energy used during the sonication is low and the time of sonication is reduced (3 min), the final temperature is not much higher than the initial temperature. Nevertheless, as confirmed by DSC analyses, Retinol CB 50 is stable up to 180 ËšC.
Once again, we thank you for your thoughtful review and look forward to submitting an improved version of our work for your consideration.